# Is microfinance associated with changes in women's well-being and children's nutrition? A systematic review and meta-analysis

Wanjiku Gichuru,[1] Shalini Ojha,[2] Sherie Smith,[3] Alan Robert Smyth,[3] Lisa Szatkowski[1]

[1]Division of Epidemiology and Public Health, University of Nottingham, Nottingham City Hospital, Nottingham, UK
[2]Division of Graduate Entry Medicine, Derby Medical School, University of Nottingham, Nottingham, UK
[3]Division of Child Health, Obstetrics and Gynecology, Queen's Medical Centre, University of Nottingham, Nottingham, UK

**Correspondence to**
Dr Shalini Ojha;
shalini.ojha@nottingham.ac.uk

## ABSTRACT

**Background** Microfinance is the provision of savings and small loans services, with no physical collateral. Most recipients are disadvantaged women. The social and health impacts of microfinance have not been comprehensively evaluated.

**Objective** To explore the impact of microfinance on contraceptive use, female empowerment and children's nutrition in South Asia, Sub-Saharan Africa and Latin America and the Caribbean.

**Design** We conducted a systematic search of published and grey literature (1990–2018), with no language restrictions. We conducted meta-analysis, where possible, to calculate pooled ORs. Where studies could not be combined, we described these qualitatively.

**Data sources** EMBASE, MEDLINE, LILACS, CENTRAL and ECONLIT were searched (1990–June 2018).

**Eligibility criteria** We included controlled trials, observational studies and panel data analyses investigating microfinance involving women and children.

**Data extraction and synthesis** Two independent reviewers extracted data and assessed risk of bias. The methodological quality of included studies was assessed using the Cochrane risk-of-bias tool for controlled trials and quasi-experimental studies and a modified Newcastle Ottawa Scale for cross-sectional surveys and analyses of panel data. Meta-analyses were conducted using STATA V.15 (StataCorp).

**Results** We included 27 studies. Microfinance was associated with a 64% increase in the number of women using contraceptives (OR 1.64, 95% CI 1.45 to 1.86). We found mixed results for the association between microfinance and intimate partner violence. Some positive changes were noted in female empowerment. Improvements in children's nutrition were noted in three studies.

**Conclusion** Microfinance has the potential to generate changes in contraceptive use, female empowerment and children's nutrition. It was not possible to compare microfinance models due to the small numbers of studies. More rigorous evidence is needed to evaluate the association between microfinance and social and health outcomes.

**PROSPERO registration number** CRD42015026018.

### Strengths and limitations of this study

► A critical evaluation of the limited evidence of the effects of microfinance on social and health outcomes.
► Encompasses all regions of the low-and-middle income countries where microfinance is most likely to impact health and well-being of vulnerable populations.
► Broad search terms used to capture all types of microfinance and a range of terminologies for the chosen outcomes.
► No language restrictions—captured all Latin American literature which is vital in the field of microfinance.
► We found few randomised controlled trials in the field and relied on the inclusion of quasi-experimental studies.

## INTRODUCTION
### Rationale
Microfinance is the provision of financial services, including savings, deposit and credit services, to the poor.[1] The term was first used in the early 1990s though schemes have been in operation in the low-income and middle-income countries since the 1970s.[2] 'Microfinance' is subtly distinct from 'microcredit,' which refers to only small loans to poor people without a savings component. Microfinance may also include provision of micro-insurance as an 'add on' to the loans and saving component. Distinct characteristics of microfinance schemes are that they are short-term, have simple application procedures and do not require loan security but instead rely on a 'collective' guarantee from an enrolled group.[3] The purpose of microfinance is that the loans should reach the poor and move them out of poverty.[4]

The financial viability of microfinance programmes may be assessed by factors such as loan size, number of loans per person

and repayment rates. One of the first studies to evaluate the economic impact of microfinance on participants was a quasi-experimental survey from Bangladesh.[5] This showed a reduction in moderate and extreme poverty and an increase in annual household expenditure of 18% among female, and 11% among male, borrowers. Institutions such as the World Bank, International Monetary Fund and the United Nations have since supported microfinance. There are currently over 3500 microfinance institutions (MFIs) providing financial support to 170 million people worldwide, mostly in South Asia, Sub-Saharan Africa (SSA), Latin America and the Caribbean (LAC).[6]

There is an emerging body of literature, including both experimental and quasi-experimental studies, looking at the social and health outcomes of microfinance programmes. In some cases, individual studies from the same region have reported contradictory results. For example, one study in Ghana demonstrated that combining microfinance and nutritional education led to improved indicators of children's nutrition in the intervention group,[7] while a study in Ethiopia failed to demonstrate any difference in nutrition status between the children of clients and non-clients.[8] The two studies used different nutritional outcome measures as well as different age limits which makes synthesis of the findings difficult. Similarly, a study from Bangladesh reported improved female empowerment 15 years later[9], but there was no significant effect in a study in Hyderabad, India.[10] Most available studies are small and have insufficient power to detect small changes in outcomes. Therefore, this systematic review brings together results from existing studies to assess whether receiving microfinance is associated with changes in women's empowerment and the well-being of their children.

## Objectives
We aimed to evaluate the impact of microfinance schemes on health and social outcomes, specifically female contraceptive use and measures of female empowerment (intimate partner violence (IPV), decision-making ability and mobility), as well as the effects on child nutrition.

## METHODS
The protocol for this review is registered with PROSPERO and is available from http://www.crd.york.ac.uk/PROSPERO (online supplementary file: Gichuru *et al* PROSPERO protocol).

## Eligibility criteria
We included all controlled trials, observational studies and analyses of panel data from South Asia, SSA and LAC[11] in women over the age of 15 and children under 5. We included quasi-experimental studies (empirical studies used to estimate the causal impact of an intervention without randomisation). In most cases, panel data were longitudinal or 'before and after' studies. We also put in

a geographical limitation to studies in countries within three World Bank regions with the highest number of low-income and middle-income countries.[12] Studies were included where the microfinance intervention comprised both savings and credit services, without physical collateral, to a poor or otherwise vulnerable population. Studies where microfinance was introduced and measured for expected change in outcome were included. Studies where an additional intervention was delivered in addition to microfinance were also included, provided that there was an intervention group where a microfinance intervention was assessed in comparison with the control group. In studies with more than one comparison group, the group without microfinance was considered as the main comparator. Studies were excluded where there were no suitable comparison data—either from a population who had not received microfinance or preintervention data from those who went on to receive microfinance.

## Patient and public involvement
There was no patient or public involvement in the design or conduct of this review. The results were presented and discussed at a dissemination workshop in Patna, Bihar.

We conducted a workshop 'Women's Empowerment and Child Health: Exploring the Impact of Rojiroti Microfinance in Poor Communities in Bihar- An Indo-UK collaboration' in Patna, India, on 22 May 2018. It was attended by more than 30 women who participate in microfinance, and a wide range of local stakeholders. The results of this review and other work were presented and discussed in this meeting, and women's views were noted to enable further research in this area.

## Outcome measures
Box 1 lists the outcome measures used to assess the impact of microfinance. The Grameen foundation proposed three variables as indicators of the social performance of microfinance[13]: female use of contraceptives, female empowerment and children's nutrition.[14–19]

The WHO considers the health and well-being of women to be tied to their ability to access healthcare and have a say in decisions related to their health.[14] Improved health status could therefore be a possible consequence and proxy indicator of female empowerment. The WHO provides some standardised measures for use in assessing the health of women in a population. These include deaths from pregnancy-related complications, uptake of contraceptives and utilisation of perinatal services.[14 15] Uptake of contraceptives is one of the measures proposed by the Grameen Foundation.[16]

Due to the broadness of the term 'female empowerment', indicators collated from definitions used by the WHO[14 15] and the UN Millennium taskforce on gender equality[16] and also from literature on social measures of female empowerment[17 19] were used to inform the selection of the three outcome measures of female empowerment used in this systematic review. These were self-reported IPV, decision-making ability and mobility.

## Box 1 Definitions of outcome measures

**Contraceptive use**
Self-reported use of any contraceptive method to prevent or plan for pregnancy.
**Female empowerment**
**Intimate partner violence (IPV):** Self-reported IPV described as physical, sexual or psychological harm by a current or former partner.[48]
**Sole decision-making ability:** Self-reported independent decision-making ability where the woman is not the head of household; including but not limited to, household expenditure, children's education or as a combined measure of empowerment as defined by individual study authors.
**Mobility:** Self-reported freedom to travel out of the village or to attend social events without the permission or accompaniment of a male relative.
**Children's nutrition**
Standard nutritional measures for children aged <5 as defined by the WHO Global Database on Child Growth and Malnutrition (WHO). Moderate undernutrition (malnutrition) was defined as a z-score $<-2$ but $>-3$ SD from the mean. Severe undernutrition (malnutrition) was defined as a z-score $<-3$ SD from the mean.
**Weight-for-age z-score (WAZ)**
**Height (or length)-for-age z-score (HAZ):** The most indicative measure of chronic undernutrition over a prolonged period leading to growth retardation known as stunting.
**Weight-for-height (or length) z-score:** Most indicative measure of acute undernutrition known as wasting. This distinguishes short children of normal weight and tall children of low weight that may not be captured by WAZ or HAZ.
**Body mass index-for-age z-score**
**Mid-upper arm circumference (MUAC)**: An absolute measure where a MUAC <11.5 cm in children 6–60 months is considered as severe acute malnutrition (wasting) and MUAC 11–12.5 cm moderate acute malnutrition.

### Information sources

EMBASE, MEDLINE, LILACS, CENTRAL and ECONLIT were searched from 1990 (when microfinance was first described)[2] to 9 September 2015. These were accessed through www.theses.com, and the references of included studies were tracked to identify other relevant papers. No language restrictions were applied. Searches were conducted using MESH headings and free text, as described in online supplementary material, supplement 1.

### Study selection, data extraction and quality assessment

Two authors (WG and LS) independently screened the titles and abstracts of retrieved studies against the study eligibility criteria. The search was updated in June 2018. For the updated search, two authors again screened the titles and abstracts (SS and SO) of the retrieved studies and two authors (SS and WG) screened the full text and extracted data, where possible. Discrepancies were resolved by discussion and duplicates removed. Retrieved studies were translated into English, where necessary, and data were extracted by the two authors independently using a standard data extraction form. The methodological quality of included studies was assessed independently

by WG and LS using the Cochrane Risk-of-Bias tool[20] for controlled trials and quasi-experimental studies and a modified Newcastle Ottawa Scale[21] for cross-sectional surveys and analyses of panel data (online supplementary material, supplement 2).

### Data synthesis and analysis

Meta-analyses were conducted using STATA V.15 (StataCorp) to pool the measures of effects from eligible studies. Where available, adjusted measures of effect were preferred over unadjusted measures. Statistical significance was set at a p value of <0.05. A random effects model was initially fitted for each meta-analysis. For studies with low heterogeneity analysis was repeated using a fixed effects model. Publication bias was assessed using funnel plots and Egger's asymmetry test (where at least five studies were available). Descriptive synthesis was carried out where studies could not be meta-analysed.

## RESULTS

### Study selection

A total of 5659 titles were identified across the three groups of outcome measures which reduced to 5298 after removal of duplicates. From these, 5023 titles were excluded as not being on microfinance as agreed mutually by two authors; 275 abstracts were subsequently screened. A total of 17 abstracts were translated for the authors to review. Each author screened the abstracts individually then came together to compare findings. The authors disagreed on 2 abstracts under contraceptive use, 4 under children's nutrition and 36 under female empowerment. These were discussed further jointly and agreed on by mutual consensus. A total of 97 progressed to full-text screening. Reference tracking identified two additional studies for full-text screening. We included 27 articles in the final review (figure 1). Seventy titles were excluded after full-text screening with reasons for exclusion outlined in online supplementary material, supplement 3. Of the 27 included articles, 4 reported on contraceptive use, 5 on children's nutrition and 18 on indicators of female empowerment. Eighteen were from South Asia, eight from SSA and one from LAC. Table 1 summarises the characteristics of the included studies.

Nature of the microfinance interventions evaluated: The most common microfinance model was group-lending as provided by formal MFIs[9 10 22–34] and community-based organisations (CBOs).[7 8 32 35–37] MFIs required clients to be women above the age of 18, own less than 0.5 decimals of land (40.4 square metres) and have at least one household member in casual employment. Self-help groups (SHGs) and CBOs had fewer eligibility criteria but with greater emphasis on accumulation of savings.[7 24 34 38–42] In some studies microfinance was coupled with additional social and health interventions.[7 25 35 36]

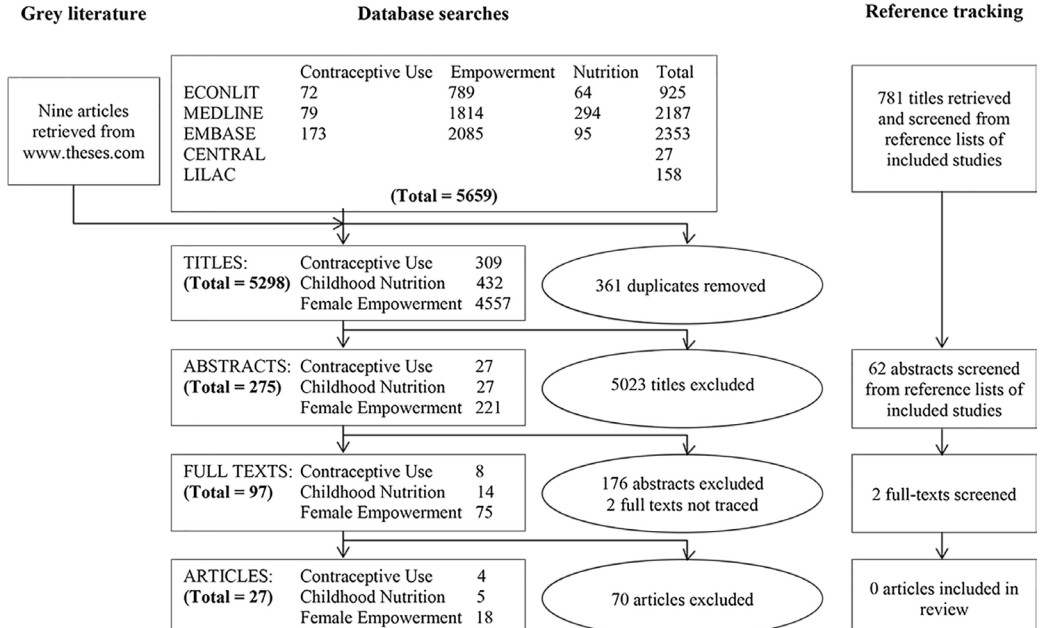

**Figure 1** Preferred Reporting Items for Systematic Reviews and Meta-Analyses flow chart.

## Findings of studies by outcome

### Contraceptive use

Four studies[5 23 24 35] evaluated the impact of microfinance on self-reported use of contraception using data from household cross-sectional surveys. One study[35] evaluated an intervention that combined microfinance with family planning education in Ethiopia. The other three studies[22–24] recruited clients from non-commercial MFIs in Bangladesh.

The impact of microfinance in the Ethiopian study was estimated at the level of the *kebele* (a cluster of villages) and showed no significant change in the proportion of married women reporting contraceptive use; individual-level estimates of the impact of microfinance were not available. A fixed-effects meta-analysis of individual-level data from the three Bangladeshi studies showed that women participating in microfinance were 64% more likely to report contraceptive use than non-clients (OR=1.64, 95% CI 1.45 to 1.86; figure 2). There was no heterogeneity between the studies which is plausible given the similarity in the average age and socioeconomic status of participants.

### Female empowerment

Seventeen studies evaluated the impact of microfinance on female empowerment. Eight were conventional cross-sectional studies,[25–28 31 38 40 43 44] three were quasi-experimental[9 30 41] and six were cluster randomised controlled trials (RCTs).[10 29 32 36 39] Twelve studies were from South Asia, three from SSA and one from LAC. These included studies evaluated different methods of empowerment.

#### Intimate partner violence

Five cross-sectional surveys[25–28 38] and one cluster RCT[36] reported this outcome. One survey[26] showed a significant

24% (95% CI 1.05 to 1.44) increase in odds of IPV among microfinance clients compared with non-clients. On the other hand, the cluster RCT[36] demonstrated a significant decrease in IPV (adjusted risk ratio 0.45, 95% CI 0.23 to 0.91) and another survey[28] similarly showed reductions among clients of the two MFIs studied (OR=0.44, 95% CI 0.28 to 0.70 and OR=0.30, 95% CI 0.18 to 0.51). Dalal *et al*[27] found that microfinance clients with secondary and higher education were 2–3 times more likely to experience IPV than comparable non-clients (p≤0.001), while wealthier clients were twice as likely to experience IPV than comparable non-clients (p≤0.001); there were no changes in exposure to IPV among the least educated and poorest groups. This finding was confirmed by Murshid *et al*[38] who also analysed the data from the same Bangladeshi Demographic Health Survey of 2007.

A meta-analysis was not conducted due to high heterogeneity ($I^2$=91.3%). This heterogeneity could have arisen because the threshold for reporting violence or the framing of the question may have differed between settings. The cluster RCT[36] was different both in design and in the add-on life skills training which may have introduced further heterogeneity. The association between IPV and microfinance is therefore inconclusive.

#### Decision-making ability

Eight studies were included for this outcome, five from South Asia[29 31 40 43 44] and three from SSA,[32 39] with four cluster RCTs,[29 32 37 39] and four cross-sectional surveys.[31 40 43 44] This measure analysed a change from not being involved in decision-making to being an active participant in household decisions. The outcome measures used were diverse and therefore unsuitable for meta-analysis. The results have been tabulated in more detail in online supplementary material, supplement 4

**Table 1** Summary of included studies

| Author, year of publication | Study design | Study setting (urban/rural, country, region) | Number of participants included in analysis | Data collection time points | Intervention provider | Services provided | Comparison group (MF) | Outcome measured | Quality assessment score |
|---|---|---|---|---|---|---|---|---|---|
| **Studies with outcome measure of contraceptive use** | | | | | | | | | |
| Desai and Tarozzi, 2011[35] | Baseline and follow-up surveys from a panel of villages: the impact of the programme was estimated using a difference-in difference approach. | Rural, Ethiopia, Sub-Saharan Africa. | 7712 women at baseline; 7949 women at follow-up. | 2003 and 2006 | CBO supported by an international NGO. | Credit and savings in group-lending model, with additional FP education. | Two comparison groups: 1. No MF or FP (used as the controls in this review). 2. FP only. | Married women aged 15–49 reporting current use of any form of contraception. | NOS 7/11 |
| Pitt and Khandker, 1996[22] | Quasi-experimental study using an econometric approach to account for non-random placement of credit programmes and unmeasured village and household attributes. | Rural, Bangladesh, South Asia. | 1731 women. | 1991, 1992 | MFI: Grameen, BRAC, BRDC. | Credit and savings in group lending model. | No MF. | Married women aged 14–50 reporting current use of any form of contraception. | NOS 4/11 |
| Steele et al, 2001[23] | Quasi-experimental study. Analysis accounted for non-random placement and self-selection by taking a random sample of women and classifying them according to their eligibility for programme membership to form target and non-target groups and considered demographic and socioeconomic variables in the analysis. | Rural, Bangladesh, South Asia. | 6456 women at baseline; 5696 women at follow-up. | 1993 and 1995 | International NGO and MFI-ASA. | Credit and savings in group lending model. | Two comparison groups: 1. No MF (used as the controls in this review). 2. Savings with no credit. | Married women reporting current use of any form of contraception. | NOS 7/11 |
| Murshid and Ely, 2017[24] | Quasi-experimental study: a logistic regression model adjusted for socioeconomic variables. | Rural, Bangladesh, South Asia. | 7325 women. | 2011 | Grameen, BRAC, ASA, Proshika, Mother's Club, BRDB or other. | Credit and savings in group lending model. | Non-participants. | Married women aged 14–50 reporting any form of contraception. | NOS 7/11 |
| **Studies with outcome measure of female empowerment** | | | | | | | | | |

Continued

**Table 1** Continued

| Author, year of publication | Study design | Study setting (urban/rural, country, region) | Number of participants included in analysis | Data collection time points | Intervention provider | Services provided | Comparison group (MF) | Outcome measured | Quality assessment score |
|---|---|---|---|---|---|---|---|---|---|
| Ahmed, 2005[25] | Data subset from a cross-sectional survey. Conduced bivariate analysis to characterise group level differences followed by a logistic regression with variables at the individual and household levels and one 'BRAC membership status' variable to account for eligibility, savings and credit. | Not reported, Bangladesh, South Asia. | 2044 women. | 1999 | MFI: BRAC. | Credit and savings in group lending model with unspecified skilled training offered to some clients. | Two comparison groups: 1. No MF (used as the controls in this review). 2. Skilled training and MF. | All women reporting either physical or verbal abuse between herself the client and her husband in the preceding 4 months. | NOS 7/11 |
| Bajracharya and Amin, 2013[26] | Cross-sectional survey: used propensity score matching to address selection bias. | Rural and urban, Bangladesh, South Asia. | 4195 women. | 2007 Demographic and Health Survey | Any MFI: Grameen, BRAC, ASA, Proshika. | Credit and savings in group lending model. | No MF. | Married women reporting any form of violence by her partner in preceding 12 months. | NOS 8/11 |
| Dalal et al, 2013[27] | Cross-sectional Survey: used $\chi^2$ test to examine difference in IPV exposure and MF and demographic variables (age, residence, education, religion and wealth index). | Rural and urban, Bangladesh, South Asia. | 4465 women. | 2007 Demographic and Health Survey | Any MFI: Grameen, BRAC, ASA, Proshika. | Credit and savings in group lending model. | No MF. | All women reporting any form of violence by her partner in preceding 12 months. | NOS 8/11 |
| Murshid et al, 2016[38] | Cross-sectional survey data were used to investigate association between MF and domestic violence with predictor variables including economic status, decision-making power and demographic variables. | Rural and urban, Bangladesh, South Asia. | 4163 women. | 2007 Demographic and Health Survey | Any MFI: Grameen, BRAC, ASA, Proshika. | Credit and savings in group lending model. | No MF. | Conflicts Tactics Scale based on the battery of questions that asked respondents whether they experienced a number of violent acts that constituted physical and sexual violence. | NOS 8/11 |

Continued

**Table 1** Continued

| Author, year of publication | Study design | Study setting (urban/rural, country, region) | Number of participants included in analysis | Data collection time points | Intervention provider | Services provided | Comparison group (MF) | Outcome measured | Quality assessment score |
|---|---|---|---|---|---|---|---|---|---|
| Pronyk et al, 2006[36] | Cluster RCT: per-protocol analysis. As only eight villages were randomised, baseline imbalances were adjusted prior to analysis. | Rural, South Africa, Sub-Saharan Africa. | 538 women (290 intervention, 248 control). | 2001, 2005 | Local NGO. | Credit and savings in group lending model with additional life skills training. | No MF. | All women reporting IPV in preceding 12 months. | Cochrane risk-of-bias: high |
| Schuler et al, 1996[28] | Cross-sectional survey. Conducted multivariate analysis using a logistic regression model with independent variables age, education, religion, whether respondent had any surviving sons or daughters, geographic region, economic level of household, respondent's contribution to family support and, exposure to credit programmes. | Rural, Bangladesh, South Asia. | 1225 women. | 1992 | MFI: Grameen and BRAC. | Credit and savings in group lending model. | No MF. | Women reporting physical beating by husband in the preceding 12 months. | NOS 7/11 |
| Angelucci et al, 2015[29] | Cluster RCT: intent-to-treat analysis on all respondents. | Rural, Mexico, Latin and Central America. | 1823 women. | 2009–2012 | MFI: Compartamos Banco. | Credit and savings in group lending model. | No MF. | Decision-making ability: participation in financial decisions and household issues by non-single women aged 18–60 who are not the only adult in their household. | Cochrane risk-of-bias: high |
| Banerjee et al, 2015[10] | Cluster RCT: intent-to-treat analysis: constructed an equally weighted average z-score of 16 social outcomes to detect any difference. | Urban, India, South Asia. | 6862 women at first follow-up; 6142 women at second follow-up. | 2005, 2010 | MFI: Spandana. | Credit and savings in group lending model. | No MF. | Index of empowerment encompassing scores across 16 domains, covering decision-making, levels of health and education expenditure and school enrolment. | Cochrane risk-of-bias: high |

**Table 1** Continued

| Author, year of publication | Study design | Study setting (urban/rural, country, region) | Number of participants included in analysis | Data collection time points | Intervention provider | Services provided | Comparison group (MF) | Outcome measured | Quality assessment score |
|---|---|---|---|---|---|---|---|---|---|
| Beaman et al, 2014[39] | Cluster RCT: intention to treat analysis. The econometric baseline characteristics and variable used in the randomisation process such as household and village characteristics. | Rural, Mali, Sub-Saharan Africa. | 5425 women. | 2009, 2012 | SHG with NGO support. | Credit and savings in SHG model. | No MF. | Decision-making ability: women's freedom to decide about food and educational expenses and take decisions about business. Index of intra-household decision-making power combining individual measures. | Cochrane risk-of-bias: high |
| Karlan et al, 2017[37] | Cluster RCT: A polled model controlling for baseline values and district was estimated by an 'intention to treat' method. | Rural, Ghana, Malawi and Uganda. | 15000 households. | Baseline 2008 to survey at endline in 2011 | Cooperative for Assistance and Relief Everywhere. | Village savings and loan associations. | No MF. | Decision-making ability: women's empowerment index capturing self-reported influence on household decisions, particularly in relation to food expenses for the household, education and healthcare expenses for the children, business expenses if the household operates a business and the women's ability to visit friends. | Cochrane risk-of-bias: high |

Continued

**Table 1** Continued

| Author, year of publication | Study design | Study setting (urban/rural, country, region) | Number of participants included in analysis | Data collection time points | Intervention provider | Services provided | Comparison group (MF) | Outcome measured | Quality assessment score |
|---|---|---|---|---|---|---|---|---|---|
| Mohindra et al, 2008[40] | Cross-sectional survey. A three step model including only SHG participation, socioeconomic characteristics and caste was examined with a goodness-of-fit test and ORs. | Rural, India, South Asia. | 928 women. | 2003 | SHG with NGO support. | Credit and savings in SHG model. | No MF. | Decision-making ability: whether women aged 18–59 reported at least one situation (of five asked) in which her husband or a male relative was the sole decision-maker. | NOS 7/11 |
| Montgomery and Weiss, 2011[43] | Cross-sectional survey: analysis accounted for income variables, consumption–expenditure variables, and household characteristics and explored differential effects on urban and rural households. | Rural and urban, Pakistan, South Asia. | 2876 women. | 2005 | Commercial MFI: Khushali. | Credit and savings in group lending model. | No MF. | Decision-making ability: women between 15 and 40 asked whether their opinion is taken into account in a series of household decisions. | NOS 7/11 |
| Pitt et al, 2003[9] | Quasi-experimental study using econometric methods similar to Pitt and Kandker.[5] | Rural, Bangladesh, South Asia. | 2074 women. | 1991/1992, 1998/1999 | MFI: Grameen, BRAC, BRDC, ASA. | Credit and savings in group lending model. | No MF. | Empowerment score combining empowerment indicators across several domains of decision-making, discussion, finance and mobility. | NOS 7/11 |
| Rahman et al, 2009[30] | Quasi-experimental cross-sectional survey. Considered age, education level, spouse's age and education level, household income, asset accumulation and locality in the analysis. | Rural and urban, Bangladesh, South Asia. | 571 recruited and analysed. | Not indicated | MFI: Grameen and BRAC. | Credit and savings in group lending model. | No MF. | Mobility index; empowerment index. | NOS 6/11 |

Continued

**Table 1** Continued

| Author, year of publication | Study design | Study setting (urban/rural, country, region) | Number of participants included in analysis | Data collection time points | Intervention provider | Services provided | Comparison group (MF) | Outcome measured | Quality assessment score |
|---|---|---|---|---|---|---|---|---|---|
| Sharif, 2004[31] | Cross-sectional survey data were used for econometric analysis with a range of socioeconomic and demographic variables. | Not reported, Bangladesh, South Asia. | 483 women. | 1997 | MFI: ASA. | Credit and savings in group lending model. | No MF. | Decision-making ability: Likert-type responses on women's extent of decision-making across six domains. | NOS 7/11 |
| Swain and Wallentin, 2009[41] | Quasi-experimental cross-sectional survey. Used the robust maximum likelihood method. | Rural and urban, India, South Asia. | 961 women. | 2000 and 2003 | SHG with MFI linkage. | Savings at group level and credit from MFI in group lending model. | No MF. | Empowerment score. | NOS 5/11 |
| Tarozzi et al, 2015[32] | Cluster RCT panel of villages data for used for an intent-to-treat analysis to identify the impact of giving access to microcredit rather than actual borrowing. | Rural, Ethiopia, Sub-Saharan Africa. | 6412 households at baseline; 6263 households at follow-up. | 2003 and 2006 | CBOs supported by international NGO. | Credit and savings in group lending model. | No MF. | Decision-making ability: fraction of decisions across 20 domains women aged 15–49 were involved in making. | Cochrane risk-of-bias:high |
| Zaman, 1999[44] | Cross-sectional survey data were used in a multivariate analysis with considerations for the number of eligible households in the village, membership length and socioeconomic differences. | Rural, Bangladesh, South Asia. | 1568 women. | 1995 | MFI: BRAC. | Credit and savings in group lending model. | No MF. | Decision-making ability. | NOS 2/11 |

**Studies with outcome measures of children's nutrition**

Continued

**Table 1** Continued

| Author, year of publication | Study design | Study setting (urban/rural, country, region) | Number of participants included in analysis | Data collection time points | Intervention provider | Services provided | Comparison group (MF) | Outcome measured | Quality assessment score |
|---|---|---|---|---|---|---|---|---|---|
| Abubakari et al, 2014[42] | Cross-sectional survey: analysis accounted for food acquisition behaviours and demographic characteristics of the households. | Rural, Ghana, Sub-Saharan Africa. | 180 children. | 2011 | Village Savings and Loans Association. | Credit and savings in SHG model. | No MF. | Anthropometric measurement of nutritional status in children <5years based on HAZ scores: >–2 well nourished; <–2 to –3 moderate malnutrition; <–3 severe malnutrition. | NOS 4/10 |
| Doocy et al, 2005[8] | Cross-sectional survey with community controls who were matched by sex and selected by proximity of residence via systematic random sampling. | Rural and urban, Ethiopia Sub-Saharan Africa. | 608 children. | 2003 | NGO: WISDOM. | Credit and savings in group lending model. | Two comparison groups: 1. No MF (used as the controls in this review). 2. New clients <1cycle of MF. | Anthropometric measurement of nutritional status in children aged 6–59 months based on MUAC: <11 cm severe malnutrition; 11–12.5 cm moderate malnutrition. | NOS 6/11 |
| Friesen et al, 2012[33] | Cross-sectional survey. Analysis included socioeconomic and demographic factors including household and maternal characteristics and child's age and sex. | Rural and urban, Ghana, Sub-Saharan Africa. | 204 children. | June–August 2011 | Local MF bank (previously with NGO support). | Credit and savings in group lending model. | No MF. | Anthropometric measurement of nutritional status in children aged 6–23 months based on proportion underweight (WAZ<–2), stunted (LAZ<–2) and wasted (WLZ<–2). | NOS 7/11 |

Continued

**Table 1** Continued

| Author, year of publication | Study design | Study setting (urban/rural, country, region) | Number of participants included in analysis | Data collection time points | Intervention provider | Services provided | Comparison group (MF) | Outcome measured | Quality assessment score |
|---|---|---|---|---|---|---|---|---|---|
| Marquis et al, 2015[7] | Quasi-experimental design with longitudinal follow-up. Bivariate analysis between anthropometric measures and explanatory variables and sensitivity analysis was performed to examine within subject variations. | Rural, Ghana, Sub-Saharan Africa. | 608 caregivers with children. | Approximately 4-monthly between April 2006 and Dec 2007 | Credit and savings association. | Credit and savings in SHG model with additional health, nutrition and entrepreneur education. | No MF. | Anthropometric measurement of nutritional status in children aged 2–5 years based on WAZ, HAZ and BAZ scores. | Cochrane risk-of-bias: high |
| Ojha et al, 2017[34] | Cluster RCT with cross-sectional follow-up and intention to treat analysis. | Rural, India, South Asia. | 1377 children. | August 2013–March 2016 | Rojiroti MF programme. | Savings and credit in peer-led SHGs. | No MF. | Anthropometric measures of children 0–5 years of age WHZ, HAZ, WAZ, MUAC. | Cochrane risk-of-bias: high |

BAZ, body mass index-for-age z-score; BRDB, Bangladesh Rural Development Board; BRDC, Bangladesh rural development corporation; CBO, community-based organisation; FP, family planning; HAZ, height (or length)-for-age z-score; IPV, intimate partner violence; LAZ, length for age z-score; MF, microfinance; MFI, microfinance institution; MUAC, mid-upper arm circumference; NGO, non-governmental organisation; NOS, Newcastle Ottawa Scale; RCT, randomised controlled trial; SHG, self-help group; WAZ, weight-for-age z-score; WHZ, weight-for-height (or length) z-score; WLZ, weight for length z-score.

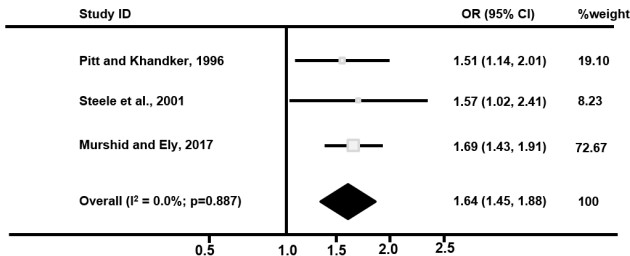

**Figure 2** Fixed effects meta-analysis of effect of microfinance participation on women reporting contraceptive use.

and include participation in financial and other household decisions (eg, children's education and healthcare). Just over half the studies[29 31 43 44] showed a slightly higher degree of participation in certain household decisions by microfinance clients compared with non-clients. The other studies did not report any statistically significant changes. The impact of microfinance on women's decision-making is therefore inconclusive.

### Freedom to travel (mobility)
In the one study that assessed mobility, non-clients were more mobile than clients in one region, but in the two other regions studied the reverse was true.[30] No formal statistical comparisons between groups were presented.

### Children's nutrition
Five studies, four from SSA[7 8 33 42] and one from India,[34] evaluated the effect of microfinance on children's nutrition. Three[8 33 42] were cross-sectional surveys, one was a quasi-experimental study with a 16-month follow-up period[7] while one was a cluster RCT.[34] Two studies[7 33] included only children between 6 and 36 months of age while the other three included children under 5 years.

Doocy et al reported that children of women non-clients were 79% more likely to be wasted than children of clients (OR=1.79 95% CI 0.87 to 3.79).[8] However, Friesen et al reported increased wasting among children of clients compared with non-clients (OR=1.15 95% CI 0.30 to 4.43).[33] Neither association was statistically significant. As the baseline group used was different and there were no raw data available, it was not possible to recalculate the ORs for pooling by meta-analysis.

One cross-sectional study found that the prevalence of malnutrition, based on height (or length)-for-age z-score (HAZ), was lower among children of microfinance clients than those of non-clients.[42] A longitudinal study measured HAZ, weight-for-age z-score (WAZ) and body mass index-for-age z-scores every 4 months for 16 months.[7] The authors demonstrated a mean difference in WAZ scores of 0.28 at 8–12 months in favour of the intervention group and significant but smaller differences at 4 months and 16 months. At 16 months, HAZ were significantly higher in the intervention group with a mean difference of 0.19 between the two groups. Meta-analysis was not possible as

**Table 2** Summary of results of the review

| Outcome | Summary of impact of microfinance |
| --- | --- |
| Use of contraception | Women participating in microfinance schemes were significantly more likely to report using contraception. |
| **Female empowerment** | |
| Intimate partner violence | Conflicting results, with some studies reporting increased and others decreased intimate partner violence in microfinance participants. |
| Decision-making ability | Most studies showed no effect but a minority showed a significant positive effect on some areas of decision-making. |
| Mobility | No statistically significant impact. |
| Overall empowerment score | Positive impact in two studies with mixed results and no change in two others. |
| Children's nutrition | Positive impact in three of five studies, with no difference found in the remaining studies. |

the studies used different statistical measures to present their results.

Ojha et al reported that in a cross-sectional survey conducted 18 months after random allocation to received immediate microfinance versus delayed microfinance (after 18 months), 0–5 years old children in the villages that received immediate microfinance had a significantly better WHZ compared with children in the villages that did not receive microfinance with a mean difference of 0.35 SD.[34] They found similar differences in WAZ, and prevalence of wasting, underweight and moderate and severe malnutrition as measured by mid-upper arm circumferences but there was no difference in HAZ or prevalence of stunting between the two groups.

### Publication Bias
A funnel plot found no evidence of publication bias in the studies that reported the impact of microfinance on IPV (Egger's test p value=0.106). The possibility of publication bias could not be assessed for the other outcomes.

### DISCUSSION
#### Summary of evidence
Table 2 summarises the impact of microfinance across the three outcome domains based on the quantitative and qualitative syntheses described above.

Seventeen of the 27 studies included in the review were from South Asia. This may limit the generalisability of the findings of this review to other geographical regions. However, this was expected as 84% of all microfinance clients are to be found in South Asia.[45] Other included studies, nine from Africa and one from Latin America, are geographically heterogeneous but catered to women of a similar economic background. These populations

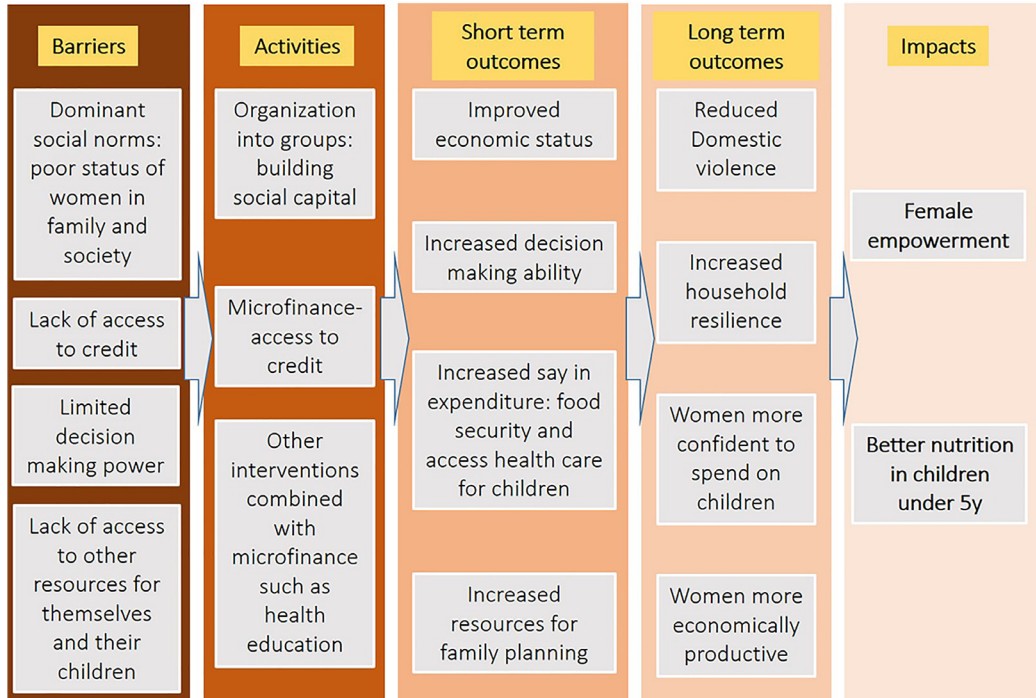

**Figure 3** Theory of change model linking microfinance to women's well-being and children's nutrition.

are potentially comparable for the purposes of a study looking at the impact of microfinance. However, it is of note that the review includes populations from a wider geographical range, with diverse political, cultural and social backgrounds.

### Proposed mechanisms

Microfinance (while primarily improving economic stability) might empower women and improve child nutrition though a number of mechanisms. A small source of income, which is available primarily to the woman in the household, could increase the 'bargaining power' of female participants, in household decision-making. Peer support and shared learning from other participants might have a similar effect. We have chosen the outcome measures most likely to reflect this increased bargaining power, including a woman's decisions about contraception and her self-reported empowerment. Furthermore, that women are often the primary household decision-makers on issues such as buying food (which will affect child nutrition) and on access to healthcare for children. These factors could interact to enable women to overcome social, cultural and economic barriers that affect their status (figure 3)

### Contraceptive use

Where individual-level data were available, the odds of reporting contraceptive use were higher in women participating in microfinance compared with those who did not. It has been argued that the women who self-select to join microfinance groups are more empowered than other women, and this may in itself increase their likelihood of using contraception.[4] However, by comparing reported use in this group before and after the intervention,[23 35]

it is possible to demonstrate a positive effect attributable to microfinance, even with an inherent empowered state.

### Markers of female empowerment
#### Intimate partner violence

Gender-related violence is known to be most commonly perpetuated by a person close to the woman, usually an intimate partner.[45] Although a reduction in IPV is one of the expected benefits of empowerment of women through microfinance, empowerment may also enable women to report more IPV, thus increasing the rate of reported IPV. One cluster RCT[36] reported a reduction in IPV among microfinance clients. However, the combined microfinance with life skills training may have resulted in an intervention group different from the standard client therefore limiting the generalisability of their findings. The authors of this study argued that their training empowered the women to reveal IPV, therefore reducing under-reporting.[36] Under-reporting of IPV is common in many studies due to its sensitive nature.[46] Studies used trained local female interviewers to limit under-reporting, but despite this, the response rate to IPV questions in one study was only 41%.[27] Furthermore, women participating in microfinance may want to only highlight positive impacts of the intervention and not reveal any IPV. This raises ethical concerns that studies may fail to detect violence where it is actually present.[46]

Studies that have reported increase in IPV linked to microfinance programmes[27] have also argued that microfinance loans may have caused more economic stress in the family leading to greater occasions for conflict. Some authors explain this as the 'status inconsistency theory' where in status differentials may lead to dysfunctional behaviour

when and individual who expects to have a higher status in a relationship is threatened by the increase in the status of another.[38] Previously, there may have been fewer conflicts as the man would have managed finances single-handedly while with empowerment, the wife becomes involved in these decisions, generating more occasions where conflict leading to IPV could occur.

## Decision-making ability

In most cases, the decision-making ability of women participating in microfinance was not significantly different from that of non-clients. However, most studies analysed women's perceived decision-making ability which may be different to their actual decision-making capability. In addition, composite indices of decision-making ability make it hard to untangle any impact of microfinance on decisions which are typically male-dominated (such as child marriage and education) and decisions which are traditionally less so (such as those related to the purchase of food).

## Children's nutrition

Three studies[8 33 34] reported a lower likelihood of severe acute malnutrition in children of women participating in microfinance compared with non-participants, including one that showed a statistically significant reduction in malnutrition.[34] Combining microfinance with nutritional education, as was the case in one study,[7] showed improvement in nutritional status in children of participating care-givers than non-participating care-givers. However, it is then difficult to isolate the specific effect of microfinance. In one SHG study[42] no attempt was made to adjust for other variables, such as household resources or education status, which may be a source of confounding.

Additionally, the inclusion of HAZ scores as a measure of nutritional status[33 42] in a cross-sectional study may be misleading. In their cluster randomised trial, Ojha *et al* report an improvement in all other indices of malnutrition other than HAZ and stunting after an 18 month period.[34] Height-for-age measures the effect of poor nutrition on the growth of a child. Growth faltering is slow in reversal and requires a longer follow-up period to detect.[47] It may be more prudent to use acute measures of malnutrition such as wasting (WHZ) which are likely to be more sensitive to change in nutritional status over shorter periods.

## Strengths and limitations

Five comprehensive databases were searched in this review, including a large economic database. The use of multiple indicators to measure women's empowerment and children's nutrition also served to broaden the search to reduce the likelihood of missing relevant articles. The selection was carried out independently by two authors without any language restrictions, particularly important given the geographical regions studied.

The models used to deliver microfinance services varied across included studies. Some combined microfinance with education on family planning,[35] life skills[36] or

health, nutrition and entrepreneurial skills[7] which made it difficult to evaluate the effect of microfinance alone. Although all interventions were taken to be similar for the purposes of this review, it was possible that the way the microfinance services were provided might have influenced the outcome. Given the small number of interventions of each type reviewed here, it is not possible to suggest a model of microfinance that is superior to others in terms of social performance.

In general, the most common source of bias in studies of the social impact of microfinance is selection bias, as participants *self-select* to either participate or not participate in the programme. Although, it may be argued that it would be difficult to randomise people to microfinance as the intervention may not be desired by all; therefore measuring effectiveness in those who did not desire it to begin with, may be problematic. While a cluster RCT might guard against selection bias, a recent study[10] highlighted the current challenge in achieving randomisation due to the widespread diffusion of microfinance in some regions of South Asia leading to difficulties in identifying unexposed control clusters. Therefore, we included non-randomised studies in this review in order to not limit the evidence considered. The non-randomised studies included dealt with self-selection bias in two main ways, using either panel data in a quasi-experimental design or propensity score matching (PSM). However, additional analysis in of one of the studies included in this review suggested that the reduction in IPV demonstrated using conventional statistical methods did not hold when PSM was used.[26]

Due to the lack of high-quality RCTs in this field, the vast majority of studies included in this study were cross-sectional. As a study design, cross-sectional studies do not provide the strongest level of evidence. Analysis of quasi-experimental and panel data studies proved difficult as there is currently no universally acceptable quality assessment tool. The use of the Cochrane risk-of-bias tool in this instance may have introduced an overestimation or underestimation of the risk of bias and, consequently, the quality assessment of the study.

There was a lack of homogeneity in the measures used to assess social performance of microfinance particularly that of decision-making ability which varied from study to study which may account for the conflicting outcomes. The average follow-up period of the studies included was 3 years. An alternative explanation for their statistically non-significant findings is that the observation period may have not been long enough to detect any change or may have missed any fleeting changes that occurred before the follow-up survey. While changes in some measures of children's malnutrition may be detectable within 3 years, changes in other outcomes requiring a shift in cultural and social norms may take much longer.

## CONCLUSIONS

In conclusion, our findings suggest that for the types of microfinance interventions assessed in this study, there may be an association between microfinance and increasing contraceptive use, improving female empowerment and better children's nutrition. However, as only 6 of 27 studies included in this review were randomised trials, any conclusions about direct causation must be guarded. However, the wide diversity in reported outcomes, study design, statistical methods and microfinance models makes it difficult to synthesise evaluation data statistically. Thus, further studies are required to evaluate the social performance of microfinance. Such studies could focus on some of the many unanswered questions such as the impact of microfinance on specific standardised measures of children's health and women's well-being such that the findings could be compared across populations. The lack of this evidence is highlighted by the paucity of good-quality studies included in this review. Other unanswered questions include the long term impact of microfinance on communities and designing studies focused on potential harm. The design of future studies requires effective and clearly described randomisation, harmonisation of appropriate outcome measures and avoidance of confounders. Incorporating evaluation methods at the onset of a microfinance programme could help address many of the weaknesses identified here. While this may not be practical in areas where microfinance is fully established, areas with an increasing number of microfinance programmes, for example, SSA, would benefit.

**Acknowledgements** We thank Magdalena Opazo Breton and Gabriella Zapata for their assistance in translating manuscripts written in Spanish and Portuguese. We thank Dr Rajeev Kamal and his team at the A N Sinha Institute of Social Sciences, Patna, India, for organising the public involvement workshop.

**Contributors** WG, LS, SO and ARS conceived and designed the study. WG and LS independently carried out the title, abstract and full text screening and quality assessment. WG conducted the meta-analyses and wrote the first draft of the paper. SS updated the search in 2018 and completed the updated title and abstract screening. The updated full-text screening was performed by SO, SS and WG. All authors critically revised subsequent drafts, and have approved the final version.

**Funding** This work was supported by the Medical Research Council [grant number MR/M021904/1], UK.

**Competing interests** None declared.

**Patient consent for publication** Not required.

**Provenance and peer review** Not commissioned; externally peer reviewed.

**Data sharing statement** This is a secondary analysis of published data. We do not hold any unpublished data from the study. Further information about the data analysis can be obtained by contacting the corresponding author.

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
