## [Reviewer comments · BMJ Open]

ARTICLE DETAILS

TITLE (PROVISIONAL)	Is microfinance associated with changes in women's wellbeing and children's nutrition? A systematic review and meta-analysis
AUTHORS	Gichuru, Wanjiku; Ojha, Shalini; Smith, Sherie; Smyth, Alan; Szatkowski, Lisa

VERSION 1 – REVIEW

REVIEWER	Caio Piza The World Bank Research Group
REVIEW RETURNED	18-Apr-2018

GENERAL COMMENTS	I see clear improvements in the current version, but still believe that critical issues are either underdeveloped or remained unaddressed. For instance, I think that a theory of change is critical and should be included. A more careful discussion of the econometric methods used by the studies included is needed. Cross sectional survey is not a method, it just say how the data is available for analysis. The same with panel data. The authors need to say that propensity score methods were used on cross-sectional data to deal with selection in observed characteristics. Panel data (or before-and-after) was used to control for selection in time-invariant unobserved characteristics. Something like that. I found only one forest plot in the appendix. I think this is not sufficient to call this study a meta-analysis. There are many other things the authors could do with the data available when forest plots and meta-regressions are not feasible. See, for instance, Grimm and Paffhausen (Labour Economics, 2015, vol. 32). Finally, I don't think they can conclude that "our findings suggest that microfinance may be associated with...". Given the eligibility criteria used by the authors to select the studies, they should say something like: "our evidence suggest that, for the types of MFIs assessed in the studies included, in the regions where they operate, there is a positive association between A and B."
--

REVIEWER	Ronak Patel Harvard Medical School, USA
REVIEW RETURNED	19-Apr-2018

GENERAL COMMENTS	I thank the authors for taking all the reviewer comments into account and responding accordingly with clarifications to the manuscript that satisfy myself and hopefully the other reviewers.
---

REVIEWER	Anjalee Kohli Georgetown University, USA
REVIEW RETURNED	19-May-2018

GENERAL COMMENTS	Introduction 1. The authors state that the objective of the review is the following: “We aimed to evaluate the impact of microfinance schemes on social outcomes, specifically female contraceptive use and measures of female empowerment (intimate partner violence, decision making ability and mobility), as well as the effects on child nutrition.” As contraceptive use is an indicator of health (that may be linked to social behaviors), the authors should consider that they are evaluating the impact of microfinance schemes on health and social outcomes. Methods 1. The authors have improved the section on outcome measures but it as written, it is still confusing for the reader to understand which measures were selected as outcome measures. It is not clear why there is discussion on WHO health indicators, repeated discussion of contraception uptake and gender pay gap, school enrollment, etc. Is this because these were assessed in the review? Or are the authors trying to make a broader argument about the relevance of the selected outcome indicators? If the latter, this seems like a discussion for either the introduction or discussion section or not the methods. 2. The selection of sole decision making ability in Table 1 should be more tightly defined – which types of decisions were the authors looking at? The authors also use the word agency in their definition of sole decision making ability – I’d like to understand whether and how agency fit in this definition? Is this a standard WHO or UN definition? If so, please cite as such. In general (and not just in this paper) I am not sure that this indicator is really focused on what is valuable or why sole decision making authority is important or stressed in these projects (as compared to joint and/or sole decision making? Results 1. The title of the decision making section should reflect the outcome of interest. Here the authors title it “decision making agency” but that is not the indicator that they looked at (or they need to correct the presentation of the outcome in the methods). 2. The authors defined empowerment in the methods and Table 1 as comprised of 3 indicators (for this paper). In the results, they describe overall empowerment as a separate indicator – can the authors please clarify in the methods what the definition of overall empowerment is and whether it was a separate outcome in the methods section including Table 1. This sentence in the results section is unclear “...based on women’s answers to questions on social and economic issues thought to have gender implications” Discussion 1. Table 3: please ensure that the wording of the outcome variables is consistent throughout the manuscript. The authors use different terms in the methods, results and discussion and this can be confusing. 2. Proposed mechanisms: This section seems more appropriate for the introduction than the conclusion. The authors state that they measured “feelings of empowerment” – please re-read this manuscript for 3. The authors state: “However, by comparing reported use in this group before and after the intervention^{29,31}, it may be possible to demonstrate any effect attributable to microfinance, even with an
---

	inherent empowered state.” Based on this analysis, are you speculating whether it is possible to draw this conclusion? In the discussion, it would be useful to demonstrate a path forward – so if you think it’s possible, what are you suggesting? Or do you think this analysis was conclusive? 4. The discussion on IPV would benefit from a look at other literature which has described economic interventions (or microfinance interventions) as inconclusive on IPV risk as this may be interacting with other dynamics in the household. I don’t have an exact reference on hand, but people have asked this question for economic programming previously and have described what they think is the rationale for this. 5. It seems like a heading for decision making is missing in the conclusion – please check. 6. In the limitations the authors argue that selection bias may be an issue in these analyses. One could also argue that it is difficult to randomize people to microfinance or not as the intervention may not be desired by all and evaluating its effectiveness amongst groups of people who do not all choose, initially, to be a part of such an intervention could be problematic.
--	---

VERSION 1 – AUTHOR RESPONSE

Reviewer(s)' Comments to Author:

Reviewer: 1

Reviewer Name: Caio Piza

Institution and Country: The World Bank Research Group Please state any competing interests: None declared.

Please leave your comments for the authors below

I see clear improvements in the current version, but still believe that critical issues are either underdeveloped or remained unaddressed. For instance, I think that a theory of change is critical and should be included.

- Thank you. We have added a Theory of change model.

A more careful discussion of the econometric methods used by the studies included is needed. Cross sectional survey is not a method, it just say how the data is available for analysis. The same with panel data. The authors need to say that propensity score methods were used on cross-sectional data to deal with selection in observed characteristics. Panel data (or before-and-after) was used to control for selection in time-invariant unobserved characteristics. Something like that.

- We have added details about the methods in Table 2.

I found only one forest plot in the appendix. I think this is not sufficient to call this study a meta-analysis. There are many other things the authors could do with the data available when forest plots and meta-regressions are not feasible. See, for instance, Grimm and Paffhausen (Labour Economics, 2015, vol. 32).

- We have used the standard meta-analysis method recommended by the Cochrane collaboration and not performed economic analyses. We have followed the PRISMA statement¹ which gives reporting guidelines for systematic reviews and meta-analyses. The guidance on the wording of the study title states: “Identify the report as a systematic review,

meta-analysis, or both." We used both systematic review meta-analysis methodology and so we have included both terms in the title.

Finally, I don't think they can conclude that "our findings suggest that microfinance may be associated with...". Given the eligibility criteria used by the authors to select the studies, they should say something like: "our evidence suggest that, for the types of MFIs assessed in the studies included, in the regions where they operate, there is a positive association between A and B."

- We have modified the conclusion as suggested by the reviewer.

Reviewer: 2

Reviewer Name: Ronak Patel

Institution and Country: Harvard Medical School, USA Please state any competing interests: None declared.

Please leave your comments for the authors below

I thank the authors for taking all the reviewer comments into account and responding accordingly with clarifications to the manuscript that satisfy myself and hopefully the other reviewers.

- Thank you.

Reviewer: 3

Reviewer Name: Anjalee Kohli

Institution and Country: Georgetown University, USA Please state any competing interests: None declared

Please leave your comments for the authors below

Introduction

1. The authors state that the objective of the review is the following: "We aimed to evaluate the impact of microfinance schemes on social outcomes, specifically female contraceptive use and measures of female empowerment (intimate partner violence, decision making ability and mobility), as well as the effects on child nutrition." As contraceptive use is an indicator of health (that may be linked to social behaviors), the authors should consider that they are evaluating the impact of microfinance schemes on health and social outcomes.

- Thank you. We have added this to the objectives.

Methods

1. The authors have improved the section on outcome measures but it as written, it is still confusing for the reader to understand which measures were selected as outcome measures. It is not clear why there is discussion on WHO health indicators, repeated discussion of contraception uptake and gender pay gap, school enrollment, etc. Is this because these were assessed in the review? Or are the authors trying to make a broader argument about the relevance of the selected outcome indicators? If the latter, this seems like a discussion for either the introduction or discussion section or not the methods.

- We have modified the "outcome measures" section within the Method to further clarify the outcomes selected. We think that the mention of the Health indicators used by WHO and UN taskforce are required to clarify why we choose these measures.

2. The selection of sole decision making ability in Table 1 should be more tightly defined – which types of decisions were the authors looking at?

- we have added a more specific definition in Table 1.

The authors also use the word agency in their definition of sole decision making ability – I'd like to understand whether and how agency fit in this definition? Is this a standard WHO or UN definition? If so, please cite as such. In general (and not just in this paper) I am not sure that this indicator is really focused on what is valuable or why sole decision making authority is important or stressed in these projects (as compared to joint and/or sole decision making?)

- We agree with the above and have replaced “agency” with “ability”

Results

1. The title of the decision making section should reflect the outcome of interest. Here the authors title it “decision making agency” but that is not the indicator that they looked at (or they need to correct the presentation of the outcome in the methods).

- This section is now headed “decision making ability”.

2. The authors defined empowerment in the methods and Table 1 as comprised of 3 indicators (for this paper). In the results, they describe overall empowerment as a separate indicator – can the authors please clarify in the methods what the definition of overall empowerment is and whether it was a separate outcome in the methods section including Table 1. This sentence in the results section is unclear “...based on women’s answers to questions on social and economic issues thought to have gender implications”

- Thank you for pointing this out. We have removed the section on Overall empowerment as we did not define this composite outcome separately in the methods.

Discussion

1. Table 3: please ensure that the wording of the outcome variables is consistent throughout the manuscript. The authors use different terms in the methods, results and discussion and this can be confusing.

- We have changed the words for consistency.

2. Proposed mechanisms: This section seems more appropriate for the introduction than the conclusion. The authors state that they measured “feelings of empowerment” – please re-read this manuscript for

- We have replaced “feeling of” with “self-reported”
- We believe this section can be in the Introduction or the Discussion and would prefer to keep it in the discussion. We have added the Theory of Change diagram here on the basis of another reviewer’s comments. However, we can move this section to the Introduction if needed.

3. The authors state: “However, by comparing reported use in this group before and after the intervention^{29,31}, it may be possible to demonstrate any effect attributable to microfinance, even with an inherent empowered state.” Based on this analysis, are you speculating whether it is possible to draw this conclusion? In the discussion, it would be useful to demonstrate a path forward – so if you think it’s possible, what are you suggesting? Or do you think this analysis was conclusive?

- We have modified this statement to clarify the meaning. We think that the results demonstrate that even within an inherently empowered group (i.e. women who choose to join microfinance groups), before-after analysis demonstrates that participation in microfinance increases the odds of reporting contraceptive use. This positive effect can therefore be attributed to participation in microfinance.

4. The discussion on IPV would benefit from a look at other literature which has described economic interventions (or microfinance interventions) as inconclusive on IPV risk as this may be interacting with other dynamics in the household. I don't have an exact reference on hand, but people have asked this question for economic programming previously and have described what they think is the rationale for this.

- Thank you. We have added this to the discussion on IPV.

5. It seems like a heading for decision making is missing in the conclusion – please check.

- This heading has now been included.

6. In the limitations the authors argue that selection bias may be an issue in these analyses. One could also argue that it is difficult to randomize people to microfinance or not as the intervention may not be desired by all and evaluating its effectiveness amongst groups of people who do not all choose, initially, to be a part of such an intervention could be problematic.

- Thank you. We have added this. However it has been possible to randomise clusters of women to early or late microfinance - for an example see Ojha S et al. Arch Dis Child 2017;102(A3).

REFERENCES

1. Moher D, Liberati A, Tetzlaff J, Altman DG. Preferred reporting items for systematic reviews and meta-analyses: the PRISMA statement. *BMJ* 2009; **339**: 332-6.

VERSION 2 – REVIEW

REVIEWER	Caio Piza The World Bank Research Group
REVIEW RETURNED	08-Aug-2018

GENERAL COMMENTS	There are minor things that need to be fixed before sending this to publication. See comments attached. I personally think that the section on study limitations can be much improved. The authors could discuss the heterogeneity of target population in the reviewed studies, the quality of the evidence in each study, the heterogeneity in studies' findings, and outline some unanswered questions. Reviewer: 1 Reviewer Name: Caio Piza Institution and Country: The World Bank Research Group Please state any competing interests: None declared. Please leave your comments for the authors below
--

	I see clear improvements in the current version, but still believe that critical issues are either underdeveloped or remained unaddressed. For instance, I think that a theory of change is critical and should be included. - Thank you. We have added a Theory of change model. A more careful discussion of the econometric methods used by the studies included is needed. Cross sectional survey is not a method, it just say how the data is available for analysis. The same with panel data. The authors need to say that propensity score methods were used on cross-sectional data to deal with selection in observed characteristics. Panel data (or before-and-after) was used to control for selection in time-invariant unobserved characteristics. Something like that. - We have added details about the methods in Table 2. Reviewer: there is too much information now that are not necessarily related with study design, but with econometric models/strategies used. Please take a look at this book or any similar that describes the methods understood as being quasi-experimental. This can be quickly fixed. I found only one forest plot in the appendix. I think this is not sufficient to call this study a meta-analysis. There are many other things the authors could do with the data available when forest plots and meta-regressions are not feasible. See, for instance, Grimm and Paffhausen (Labour Economics, 2015, vol. 32). - We have used the standard meta-analysis method recommended by the Cochrane collaboration and not performed economic analyses. We have followed the PRISMA statement¹ which gives reporting guidelines for systematic reviews and meta-analyses. The guidance on the wording of the study title states: "Identify the report as a systematic review, meta-analysis, or both." We used both systematic review meta-analysis methodology and so we have included both terms in the title. Reviewer: Meta-analysis are not always present in systematic reviews, either because the number of studies is too low, and/or the interventions covered too heterogeneous so that authors don't feel comfortable in pooling them together, and/or because reviewed studies don't report all statistics needed to computation of standardized measures, and/or even because the reviewed studies report on different outcomes. In my view, this is a systematic review, but not a meta-analysis because the authors don't use forest plots and/or meta-regression techniques to summarize the findings across studies. I think that many readers will get frustrated with a study that doesn't do what the title suggests. Finally, I don't think they can conclude that "our findings suggest that microfinance may be associated with...". Given the eligibility criteria used by the authors to select the studies, they should say something like: "our evidence suggest that, for the types of MFIs assessed in the studies included, in the regions where they operate, there is a positive association between A and B." - We have modified the conclusion as suggested by the reviewer. Reviewer: The sentence in the abstract needs to be modified too.
--	---

REVIEWER	Ronak Patel
	Harvard Medical School and Harvard Humanitarian Initiative
REVIEW RETURNED	07-Aug-2018

GENERAL COMMENTS	The authors had satisfied my previous concerns in the second version of this manuscript but I had failed to account for an updated search as the editor pointed out and this was a very welcome revision. I have some disagreements with the need for a theory of change and other clarifications from other reviewers but I do not think they alter the overall assessment of this study and it is suitable for publication.
---

VERSION 2 – AUTHOR RESPONSE

Reviewer: 1

Reviewer Name: Caio Piza

Institution and Country: The World Bank Research Group

Please state any competing interests or state 'None declared': None declared

Please leave your comments for the authors below There are minor things that need to be fixed before sending this to publication. See comments attached.

I personally think that the section on study limitations can be much improved. The authors could discuss the heterogeneity of target population in the reviewed studies, the quality of the evidence in each study, the heterogeneity in studies' findings, and outline some unanswered questions.

Response

- We have commended on the diversity of the included population on Page 17. We have now further explained this heterogeneity. (The page numbers refer to the page in the submission with marker changes)

“Seventeen of the 27 studies included in the review were from South Asia. This may limit the generalisability of the findings of this review to other geographical regions. However, this was expected as 84% of all microfinance clients are to be found in South Asia⁴⁶. Other included studies, nine from Africa and one from Latin America, are geographically heterogeneous but catered to women of a similar economic background. These populations are potentially comparable for the purposes of a study looking at the impact of microfinance. However, it is of note that the review includes populations from a wider geographical range, with diverse political, cultural and social backgrounds.”

- We have added a section on quality of studies included on Page 20.

“Due to the lack of high quality randomised controlled trials in this field, the vast majority of studies included in this study were cross-sectional. As a study design, cross-sectional studies do not provide the strongest level of evidence. Analysis of quasi-experimental and panel data studies proved difficult as there is currently no universally acceptable quality assessment tool. The use of the Cochrane Risk-of-Bias tool in this instance may have introduced an over-or under-estimation of the risk of bias and, consequently, the quality assessment of the study.”

- We have further acknowledge the heterogeneity in the study findings by adding to the section on limitations on Page 20.

“There was a lack of homogeneity in the measures used to assess social performance of microfinance particularly that of decision making ability which varied from study to study which may account for the conflicting outcomes.”

- We think we have covered the lack of evidence and unanswered questions throughout the review. We have further added to the conclusion (on Page 21)

"Such studies could focus on some of the many unanswered questions such as the impact of microfinance on specific standardised measures of children's health and women's wellbeing such that

the findings could be compared across populations. The lack of this evidence is highlighted by the paucity of good quality studies included in this review. Other unanswered questions include the long term impact of microfinance on communities and designing studies focused on potential harm."

Reviewer: 2

Reviewer Name: Ronak Patel

Institution and Country: Harvard Medical School and Harvard Humanitarian Initiative

Please state any competing interests or state 'None declared': None declared.

Please leave your comments for the authors below

The authors had satisfied my previous concerns in the second version of this manuscript but I had failed to account for an updated search as the editor pointed out and this was a very welcome revision. I have some disagreements with the need for a theory of change and other clarifications from other reviewers but I do not think they alter the overall assessment of this study and it is suitable for publication.

Response:

- Thank you for your comments.